# FINDING SYMMETRY IN NEURAL NETWORK PARAMETER SPACES

## ABSTRACT

Parameter space symmetries, or loss-invariant transformations, are important for understanding neural networks' loss landscape, training dynamics, and generalization. However, identifying the full set of these symmetries remains a challenge. In this paper, we formalize data-dependent parameter symmetries and derive their infinitesimal form, which enables an automated approach to discover symmetry across different architectures. Our framework systematically uncovers parameter symmetries, including previously unknown ones. We also prove that symmetries in smaller subnetworks can extend to larger networks, enabling direct generalization of discovered symmetries to more complex models.

## 1 INTRODUCTION

Parameter space symmetry, or loss-invariant transformation of parameters, influences various aspects of deep learning theory. Continuous symmetry connects groups to their orbits, revealing topological properties such as the dimension (Zhao et al., 2023b) and connectedness (Zhao et al., 2023a) of the minimum. Parameter symmetry also influences training dynamics through the associated conserved quantities of gradient flow (Kunin et al., 2021) and by steering stochastic gradient descent towards certain favored solutions (Ziyin, 2024). Additionally, symmetry provides a tool to perform optimization within a loss level set, with successful applications in accelerating optimization (Armenta et al., 2023; Zhao et al., 2022) and improving generalization (Zhao et al., 2024). Other applications of parameter symmetry include model compression (Ganev et al., 2022; Sourek et al., 2021), efficient sampling in Bayesian neural networks (Wiese et al., 2023), and equivariant architectures for weight space learning (Navon et al., 2023; Zhou et al., 2023).

Despite the wide range of applications, our knowledge of parameter space symmetries remains limited. In particular, known symmetries often cannot account for all loss-invariant parameter transformations. While several frameworks have been developed to unify known symmetries, whether the symmetries in current literature are complete remains an open question. The lack of a systematic approach necessitates deriving symmetries from scratch for each new architecture, creating barriers for broader application of parameter symmetries.

In this paper, we present an automated approach for discovering symmetry groups and their group actions in the parameter space of neural networks. To define the search space, we formalize data-dependent symmetries and derive their infinitesimal version, which simplifies the automatic discovery architecture. Additionally, we learn the action maps directly using a neural network, enabling the discovery of nonlinear group actions. By including data-dependent and nonlinear group actions, our framework is capable of capturing a broader range of symmetries than previously considered.

While directly searching for symmetries in modern architectures with billions of parameters is prohibitively expensive, we show that large networks often inherit symmetries from their components or subnetworks. Analyzing these smaller networks provides an efficient and scalable way to uncover many symmetries in larger architectures. By extending the symmetries of small networks to their larger counterparts, our method sidesteps the complexity of handling high-dimensional parameter spaces, reducing computational cost. This approach also provides a framework for leveraging small-scale symmetries to better understand the structure of more complex architectures.

In summary, our main contributions are:

- A formal definition of data-dependent parameter symmetries and their infinitesimal form.
- An approach to identify symmetries in the parameter space of large networks from known symmetries in smaller subnetworks.
- A framework for automated discovery of parameter symmetries across diverse neural network architectures.
- Preliminary evidence of previously unknown symmetries that are data-dependent or act on non-contiguous layers.

## 2 RELATED WORK

**Parameter space symmetry.** Parameter symmetries are loss-invariant transformations on neural network parameters, often in the form of group actions. Symmetry often arises from equivariance of common activation functions (Godfrey et al., 2022), including invertible linear transformations in linear networks, rescaling in homogeneous networks (Badrinarayanan et al., 2015; Du et al., 2018; Petzka et al., 2020), radial rescaling in radial neural networks (Ganev et al., 2022), and translation in softmax and scaling in batchnorm functions (Kunin et al., 2021). In tanh neural networks (Chen et al., 1993), only permutation and sign flip symmetries preserve the loss function. ReLU networks, however, possess symmetries beyond the well-known rescaling (Grigsby et al., 2023). The existence and number of symmetries in most other architectures remain an open question.

**Data-dependent symmetry.** While the above symmetries leave the loss unchanged on all data, a relaxed definition, data-dependent symmetry, only requires loss invariance on a subset of data. Zhao et al. (2023b) found examples of such symmetries with nontrivial data dependency, although these symmetries are complicated, limited to minibatches of size one, and difficult to generalize across different architectures. This motivates an automated symmetry discovery framework, which, in principle, can find symmetries of arbitrary form in arbitrary architectures. The concept of a symmetry dependent on data has also appeared in adjacent fields. For example, Moskalev et al. (2023) observe that learned data invariance in neural networks is strongly conditioned on data and breaks under data distribution drift; Sonoda et al. (2023) define a joint group action on data and parameters as part of a new proof of universal approximation theory.

**Discovering and measuring symmetry.** Various work explores learning continuous symmetries by identifying generators of Lie groups (Krippendorf & Syvaeri, 2020; Moskalev et al., 2022; Dehmamy et al., 2021; Yang et al., 2023b; Gabel et al., 2023), including cases with nonlinear group actions (Yang et al., 2023a; Shaw et al., 2024). We build on this approach to discover data-dependent group action in high-dimensional parameter spaces. While learning discrete symmetry (Zhou et al., 2021; Karjol et al., 2024) and distributions of symmetry (Benton et al., 2020; Romero & Lohit, 2022; Urbano & Romero, 2023) are also relevant, they are not the primary focus of this paper.

Extracted symmetry is often evaluated locally, by measuring function changes under infinitesimal symmetry transformations (Gruver et al., 2022) or by comparing tangent spaces of orbits under the learned group and the true symmetry group (Portilheiro, 2023). We adopt the local invariance of loss functions under symmetry transformation, similar to that defined in (Gruver et al., 2022; Moskalev et al., 2022), as the minimization objective in learning data-dependent group actions.

## 3 PARAMETER SPACE SYMMETRY

In this section, we provide a formal definition for data-dependent parameter symmetries. We then derive an alternative definition using Lie algebras, which is used to construct an automated framework for discovering parameter space symmetries in Section 5. Lastly, we provide examples of symmetries in common neural networks.

### 3.1 DATA-DEPENDENT GROUP ACTION AND SYMMETRY

Let $\Theta$ be the space of parameters and $\mathcal{D}$ be the space of data. In this paper, we consider loss functions of the form $L : \Theta \times \mathcal{D} \to \mathbb{R}$, which map parameters and a single data point to a real number. By

abuse of notation, we allow $L$ to simultaneously process multiple data points. Specifically, we sometimes define $L : \Theta \times \mathcal{D}^d \to \mathbb{R}^d$ for $d \in \mathbb{N}$ data points.

Let $G$ be a group. Consider a map $a$, which defines a map for every data batch of size $d \in \mathbb{Z}^+$:

$$a : \mathcal{D}^d \to (G \times \Theta \to \Theta)$$
$$X \mapsto (a_X : g, \theta \mapsto \theta'). \tag{1}$$

The map $a$ is a *generalized group action* on $\Theta$ if $a_X$ is a group action for every data batch $X$, meaning that it satisfies the following axioms:

$$\text{identity:} \quad a_X(I, \theta) = \theta, \qquad \forall X \in \mathcal{D}^d, \ \ \forall \theta \in \Theta.$$
$$\text{associative law:} \quad a_X(g_2, a_X(g_1, \theta)) = a_X(g_2 g_1, \theta),$$
$$\forall g_1, g_2 \in G, \ \ \forall X \in \mathcal{D}^d, \ \ \forall \theta \in \Theta.$$

We introduce our first definition to formalize data-dependent symmetry. A group action $a$ is *parameter space symmetry of $L$* if it additionally satisfies

$$\text{loss invariance:} \quad L(a_X(g, \theta), X) = L(\theta, X),$$
$$\forall g \in G, \ \ \forall X \in \mathcal{D}^d, \ \ \forall \theta \in \Theta.$$

A function $L$ has a *$G$-symmetry* if there exists a loss-invariant group action $a$. We refer to $G$ as a symmetry group of $L$. Additionally, the action $a$ is termed a *data-dependent group action* or symmetry if the map (1) has a non-trivial dependency on $X$. That is, $a$ is data-dependent if there exists $X_1, X_2 \in \mathcal{D}^d$, such that $a_{X_1} \neq a_{X_2}$.

## 3.2 Infinitesimal Symmetry

Next, we derive an infinitesimal version of parameter space symmetries. For the automatic symmetry discovery framework in Section 5, these definitions allow us to learn the group elements and actions without computing the matrix exponential, which is expensive, during training. Proofs and additional examples can be found in Appendix A.

In this paper, we restrict the symmetry group $G$ to be a linear group. That is, we assume there is a faithful representation $\rho : G \to \mathrm{GL}(n)$. The corresponding Lie algebra representation $d\rho : \mathfrak{g} \to \mathfrak{gl}(n)$ is the differential of $\rho$, mapping elements of the Lie algebra $\mathfrak{g}$ of $G$ to the Lie algebra $\mathfrak{gl}(n)$ of $GL(n)$. If $G$ is a subgroup of $\mathrm{GL}(n)$, then $\rho$ is the inclusion map, and consequently, $d\rho$ is the inclusion of $\mathfrak{g}$ into $\mathfrak{gl}(n)$.

The following theorem shows that the derivative of the loss function $L$ with respect to the parameters $\theta$ vanishes in the directions generated by the symmetry group's infinitesimal transformations. In other words, the loss function is invariant to small changes along these symmetric directions in parameter space.

**Theorem 3.1** (Infinitesimal version of loss invariance). *Let $a : \mathcal{D}^d \to (G \times \Theta \to \Theta)$ be a parameter space symmetry of a loss function $L : \Theta \times \mathcal{D}^d \to \mathbb{R}^d$. Assume that $L$ and $a_X$ is differentiable at all $X \in \mathcal{D}^d$. Let $D_\theta L|_{\theta, X} \in \mathbb{R}^{d \times \dim(\Theta)}$ be the derivative of $L$ with respect to $\theta$, and $D_g a_X|_{I, \theta} \in \mathbb{R}^{\dim(\Theta) \times \dim \mathfrak{g}}$ be the derivative of $a_X(g, \theta)$ with respect to $g$. Then, for all $\theta \in \Theta$, $X \in \mathcal{D}^d$, and $h \in \mathfrak{g}$,*

$$D_\theta L|_{\theta, X} \ D_g a_X|_{I, \theta}(h) = 0. \tag{2}$$

*Proof sketch.* Consider a smooth curve $\gamma(t) = a_X(\exp(ht), \theta)$ in $\Theta$, where $h \in \mathfrak{g}$ and $t \in \mathbb{R}$. Then, since $L$ is invariant under $a$, $L(\gamma(t), X) = L(\theta, X), \forall t \in \mathbb{R}$. The result follows from differentiating both sides with respect to $t$ at $t = 0$ and applying the chain rule. $\square$

Equation (2) states that the gradient of the loss function $L$ with respect to the parameters $\theta$ is orthogonal to the directions in parameter space generated by the infinitesimal action $D_g a_X|_{I, \theta}(h)$. This orthogonality implies that moving along these symmetric directions does not change the loss to first order, reflecting the invariance of $L$ under the group action. Figure 1 illustrates relevant directions.

Similar to the infinitesimal formulation of loss invariance, we provide an infinitesimal version of the associativity law for smooth group actions on parameter space. While Theorem 3.1 shows how local transformations in the Lie algebra preserve the value of the loss function, associativity ensures that applying successive group transformations is consistent with their composition in the group. The theorem below shows how associativity manifests at the infinitesimal level through the derivatives of the action map $a_X$.

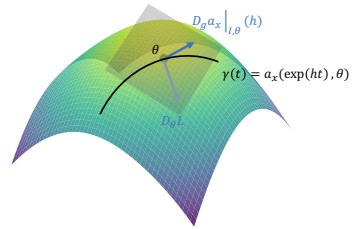

**Theorem 3.2** (Infinitesimal version of associativity). *Let $G$ be a Lie group, and let $a : \mathcal{D}^d \to (G \times \Theta \to \Theta)$ be a smooth map that satisfies the associative law. Then, for all $\theta \in \Theta$, $X \in \mathcal{D}^d$, and $h_1, h_2 \in \mathfrak{g}$,*

$$D_\theta a_X\big|_{I,\theta}\big(D_g a_X\big|_{I,\theta}(h_1)\big) + D_g a_X\big|_{I,\theta}(h_2)$$
$$= D_g a_X\big|_{I,\theta}(h_1 + h_2).$$

Figure 1: Illustration of an infinitesimal action $D_g a_X\big|_{I,\theta}(h)$ and loss gradient $D_\theta L$ at a point $\theta$ in the parameter space. The infinitesimal action is in the tangent space of the loss level set at $\theta$.

Theorem 3.2 captures how the composition of two infinitesimal actions generated by $h_1, h_2 \in \mathfrak{g}$ translates to a single infinitesimal action generated by their sum. When the group action is linear, $D_\theta a_X$ reduces to identity, and the infinitesimal action preserves the additive structure of the Lie algebra. When the group action is not linear, $D_\theta a_X$ introduces a correction term that modifies how the infinitesimal actions combine.

### 3.3 EXAMPLES

Below are examples of parameter space symmetry and their infinitesimal formulations. To facilitates calculation and interpretation, we express them in coordinate form. Assuming that $\Theta = \mathbb{R}^n$, then for a single data point ($d = 1$), we can write (2) in coordinates as

$$\sum_{i=1}^{n} \sum_{k=1}^{dim(\mathfrak{g})} \frac{\partial L}{\partial \theta_i} \left( D_g a_X\big|_{I,\theta} \right)_{ik} h_k = 0. \tag{3}$$

#### 3.3.1 LINEAR ACTION OF MATRIX GROUPS

When $\Theta = \mathbb{R}^n$ and $G$ is a subgroup of $\mathrm{GL}(n)$ with a linear, data-independent symmetry $a_x(g, \theta) = g\theta$ for all $x \in X$, this coordinate form of loss invariance reduces to the equation in Theorem 3.1 in Moskalev et al. (2022). With $(D_g a)_{ijk} = \frac{\partial a_i}{\partial g_{jk}} = \delta_{ij}\theta_k$, Equation (3) becomes

$$\sum_{i=1}^{n} \sum_{k=1}^{n} \frac{\partial L}{\partial \theta_i} \theta_k h_{ik} = 0.$$

Our symmetry acts on parameters instead of data, but otherwise this matches Theorem 3.1 in Moskalev et al. (2022).

#### 3.3.2 HOMOGENEOUS TWO-LAYER NEURAL NETWORK

We consider a homogeneous two-layer neural network with scalar weights for simplicity. Let parameter space $\Theta = \mathbb{R}^2$ and data space $X \in \mathbb{R}$. Consider the loss function

$$L : \Theta \times X \to \mathbb{R}, (w_1, w_2), x \mapsto w_2 \sigma(w_1 x)$$

with a homogeneous activation function $\sigma : \mathbb{R} \to \mathbb{R}$, i.e. $\sigma(\alpha x) = \alpha^c x$ for all $\alpha \in \mathbb{R}_{>0}$ and $x \in \mathbb{R}$, for some $c > 0$.

Let $G = (\mathbb{R}^\times, \times)$, and $\rho : G \to \mathrm{GL}_2, \alpha \mapsto \begin{pmatrix} \alpha & 0 \\ 0 & \alpha^{-c} \end{pmatrix}$. Then $a : \mathrm{GL}(2) \times \mathbb{R}^2 \to$

$\mathbb{R}^2, \left( \rho(g), \begin{pmatrix} w_1 \\ w_2 \end{pmatrix} \right) \mapsto \rho(g) \begin{pmatrix} w_1 \\ w_2 \end{pmatrix}$ is a symmetry of $L$.

# 4 BUILDING NEW SYMMETRIES FROM KNOWN ONES

One way to identify symmetries in a large network is by examining its components or subnetworks. Despite often having billions of parameters, neural networks typically consist of a limited set of function families, such as fully connected layers, attention mechanisms, and activation functions. This modular view suggests a mechanism by which symmetries in networks with fewer layers might extend to those in deeper networks. Additionally, within similar types of networks, it may be possible to extrapolate symmetries found in narrower layers to wider ones.

By focusing on symmetries in small architectures and using them to infer symmetries in larger ones, we circumvent the complexity associated with direct handling of high-dimensional parameter spaces. This approach not only simplifies the discovery of symmetries in large-scale networks but also provides a systematic method for using symmetries in smaller subnetworks to understand those in more extensive architectures. The subnetwork perspective has appeared in prevoius studies of the critical points in multilayer perceptrons Fukumizu & Amari (2000); Şimşek et al. (2021). Proposition 4.1 generalizes this observation to subnetworks in arbitrary architectures and formalizes how to obtain symmetries of large networks from the symmetries of small ones.

If a loss function $L$ depends on a subset of the parameters solely through a subnetwork $f$, then any symmetry of $f$ will also preserve $L$:

**Proposition 4.1.** *Let $L : \Theta \times \mathcal{D}^d \to \mathbb{R}^d$ where the parameter space $\Theta$ is a product space $\Theta = \Theta_1 \times \Theta_2$. Suppose for some spaces $S$ and $T$, there exist functions $h : \Theta_1 \times \mathcal{D}^d \to S$, $f : \Theta_2 \times S \to T$ and $j : (\Theta_1 \times T) \times \mathcal{D}^d \to \mathbb{R}^d$, such that for every $\theta = (\theta_1, \theta_2) \in \Theta$ and $X \in \mathcal{D}^d$, $L(\theta, X) = j\big((\theta_1, f(\theta_2, h(\theta_1, X))), X\big)$. If $a : S \to (G \times \Theta_2 \to \Theta_2)$ is a G-symmetry of $f$, then there is an induced G-symmetry of $L$, $a' : \mathcal{D}^d \to (G \times \Theta \to \Theta)$, defined by $a'_X(g, (\theta_1, \theta_2)) = \big(\theta_1, a_{h(\theta_1, X)}(g, \theta_2)\big)$.*

The relationship between the functions in the proposition is described by the commutative diagram below, where $p_1 : \Theta \to \Theta_1$, $p_2 : \Theta \to \Theta_2$ are projections onto $\Theta_1$ and $\Theta_2$, $\text{id}_1 : \Theta_1 \to \Theta_1$ and $\text{id}_2 : \Theta_2 \to \Theta_2$ are identity maps, and $X \in \mathcal{D}^d$ represents a batch of data. Space $S$ and $T$ can be interpreted as intermediate feature spaces in the neural network. When $L$ can be decomposed in this way, the function $h$ does not depend on $\Theta_2$, and the function $j$ depends on $\Theta_2$ only through the output of $f$. This effectively confines $L$'s dependency on $\Theta_2$ to the transformation defined by $f$, ensuring that any transformation on $\Theta_2$ not altering the output of $f$ will not affect the output of $L$. Consequently, symmetries identified in the smaller network $f$ can be extrapolated to the larger network $L$.

$$
\begin{array}{ccccc}
\Theta & \xrightarrow{\hspace{4cm} L(\cdot, X) \hspace{4cm}} & & & \mathbb{R}^d \\
{\scriptstyle p_1 \times p_2 \times p_1}\big\downarrow & & & & \big\uparrow{\scriptstyle j(\cdot, X)} \\
\Theta_1 \times \Theta_2 \times \Theta_1 & \xrightarrow{\text{id}_1 \times \text{id}_2 \times h(\cdot, X)} & \Theta_1 \times \Theta_2 \times S & \xrightarrow{\text{id}_1 \times f(\cdot, \cdot)} & \Theta_1 \times T
\end{array}
$$

We apply Proposition 4.1 to construct symmetries in larger networks from those in smaller ones in the next two corollaries. Specifically, we show that some symmetries are preserved as networks scale up through increasing the dimensionality of a layer or adding additional layers.

The first corollary describes how symmetries identified in narrower networks also apply to wider networks. A function $\sigma : \mathbb{R}^{h \times k} \to \mathbb{R}^{h \times k}$ is row-wise if, for any matrix $A \in \mathbb{R}^{h \times k}$ with rows $\{a_i \in \mathbb{R}^k\}_{i=1}^h$, the output matrix $\sigma(A)$ has rows $\{\sigma_{row}(a_i) \in \mathbb{R}^k\}_{i=1}^h$, where $\sigma_{row} : \mathbb{R}^k \to \mathbb{R}^k$ applies independently on each row of $A$. Element-wise functions are a special case of row-wise functions. For fully connected networks with row-wise activation functions, identifying a symmetry in one architecture suggests that the same symmetry will apply to wider versions of that architecture.

**Corollary 4.2.** *Consider a network parameter space $\Theta(m, h, n) = \mathbb{R}^{m \times h} \times \mathbb{R}^{h \times n}$ and data space $\mathcal{D}(n, k) = \mathbb{R}^{n \times k}$. Let $\sigma : \mathbb{R}^{h \times k} \to \mathbb{R}^{h \times k}$ be a row-wise function. Consider a function $L_{mnhk} : \Theta(m, h, n) \times \mathcal{D}(n, k) \to \mathbb{R}^{m \times k}$, defined as $L_{mnhk}((U, V), X) = U\sigma(VX)$ for $U \in \mathbb{R}^{m \times h}$, $V \in \mathbb{R}^{h \times n}$, and $X \in \mathbb{R}^{n \times k}$. If there is a G-symmetry of $L_{mnhk}$, then there is a G-symmetry of $L_{mnh'k}$ with any $h' > h$.*

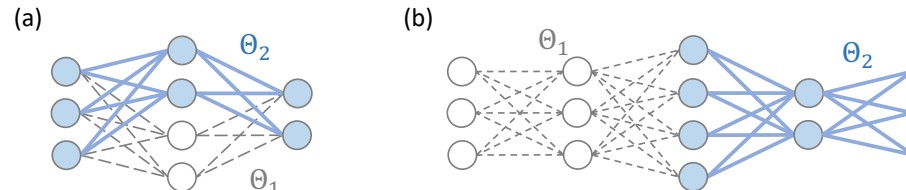

Figure 2: If a large network contains substructures with known symmetry, we can infer the same symmetry for the large network. (a) Symmetry from narrower networks. (b) Symmetry from shallower networks.

The next corollary shows that symmetries of a subset of layers are also symmetries in the entire network.

**Corollary 4.3.** *Let $\Theta = \Theta_1 \times ... \times \Theta_l$ be a parameter space. Consider a list of spaces $V_0 = \mathcal{D}^d$, $V_l = \mathbb{R}^d$, and $V_1, ..., V_{l-1}$. Let $L : \Theta \times \mathcal{D}^d \to \mathbb{R}^d$ be a function defined recursively by $\{L_i\}_{i=1}^l$ with $L_i : \Theta_i \times V_{i-1} \to V_i$, such that $L = \phi_l$ where $\phi_i = L_i(\theta_i, \phi_{i-1}) \in V_i$ and $\phi_0 = X$. If for some $1 \le i \le l$, $L_i$ has a $G$-symmetry, then $L$ has a $G$-symmetry.*

Both corollaries can be proved by factoring the parameter space and defining corresponding functions that compose to $L$, before applying Proposition 4.1. The explicit forms of $h$, $f$, and $j$ are deferred to Appendix B. Figure 2 shows the subset of parameters ($\Theta_2$) that the symmetry applies to in the corollaries. These are the subnetworks where symmetries are assumed to be known and which the larger network inherits.

Note that this approach does not explore the emergence of new, more complex symmetries that may arise as the neural network scale up in size. Notably, there are cases where there exists a $G$ symmetry over its input space, but group actions on individual layers are not loss-invariant (Kvinge et al. (2022)). Nevertheless, studying smaller and simpler networks remains a effective strategy to obtain a significant number of symmetries in larger networks, and is a first step in characterizing the complete set of symmetries in modern architectures.

In addition to obtaining symmetries from those in smaller networks, we can also get symmetries for a loss function over data batches with a certain size, if we know there is a symmetry for this function over larger data batches. Concretely, if there exists a group action that preserves loss for all data batches of size $d \in \mathbb{Z}^+$, then that group action preserves loss for all data batches of size $d' < d$.

**Proposition 4.4.** *Let $L_d : \Theta \times \mathcal{D}^d \to \mathbb{R}^d$ be a function that is applied pointwise on each of $d$ data points in a data batch. If $L_d$ admits a $G$-symmetry, then $L_{d'}$ admits a $G$-symmetry for all $d' < d$.*

## 5 AUTOMATIC DISCOVERY OF PARAMETER SYMMETRIES

Using the infinitesimal symmetry derived in Section 3.2, we construct an automated framework for discovering parameter space symmetries. Formulating symmetries in the infinitesimal form makes them easier to learn using an automatic framework, as it defines a set of local conditions for a function to be a symmetry.

### 5.1 ENFORCING LOSS INVARIANCE AND GROUP AXIOMS

Given a function $L$, our goal is to find a symmetry $a$ and a set of Lie algebra elements $h$ corresponding to a symmetry group of $L$. We parameterize $a$ using a neural network with learnable parameters, and set $h$ to be learnable as well. We define the following loss terms that quantify the deviation from loss invariance and the group axioms (identity and associativity law):

$$\mathcal{L}_{\text{invariance}} = \mathbb{E}_{x,\theta} |D_\theta L|_{\theta,X} \circ D_g a_X|_{I,\theta}(h)|$$

$$\mathcal{L}_{\text{id}} = \mathbb{E}_{x,\theta} \|a_x(I, \theta) - \theta\|_2$$

$$\mathcal{L}_{\text{assoc}} = D_\theta a_X\big|_{I,\theta}\big(D_g a_X\big|_{I,\theta}(h_1)\big)$$
$$+ D_g a_X\big|_{I,\theta}(h_2) - D_g a_X\big|_{I,\theta}(h_1 + h_2)$$

The three loss terms bias the action towards being loss-invariant, preserving identity, and satisfying the associativity property. By minimizing $\mathcal{L}_{\text{Lie\_deriv}}$, we ensure that the learned symmetry $a$ and the Lie algebra element $h$ satisfy the infinitesimal symmetry condition (Theorem 3.1). Minimizing $\mathcal{L}_{\text{id}}$ enforces the identity axiom, ensuring that the action of the identity element leaves the parameters unchanged. Minimizing $\mathcal{L}_{\text{assoc}}$ enforces the associative axiom (derivation in Appendix A.2).

By focusing on the Lie algebras, we enforce the loss invariance and group structure at the infinitesimal level. This formulation allows us to avoid computing exponential maps.

## 5.2 Regularizations

To prevent the learned group action from becoming trivial, we encourage the infinitesimal action to be nonzero. On the other hand, we do not want it to grow infinitely large for training stability. Therefore, in implementation, we include the following regularization term to encourage the norm of the infinitesimal action to be around a fixed positive real number $\beta$:

$$\mathcal{L}_{\text{reg\_id}} = \min_{a,h} \mathbb{E}_\theta | \beta - \| D_g a_X |_{I,\theta}(h) \| |.$$

When learning multiple generators simultaneously, we want them to be orthogonal. Following Yang et al. (2023b), we do this by including the following cosine similarity between each pair of the $k$ generators in the loss function:

$$\mathcal{L}_{\text{reg\_h\_orth}} = \sum_{1 \leq i < j \leq k} \frac{h_i \cdot h_j}{\|h_i\|\|h_j\|}.$$

Finally, we encourage sparsity of $h$ for easier interpretation, with the regularization term

$$\mathcal{L}_{\text{reg\_h\_sparse}} = \sum_{k,j} |h_{kj}|.$$

The final training objective is a weighted average of symmetry loss and regularizations, with hyper-parameters $\gamma_1, ..., \gamma_6 \in \mathbb{R}^+$:

$$\min_{h,a} (\gamma_1 \mathcal{L}_{\text{invariance}} + \gamma_2 \mathcal{L}_{\text{id}} + \gamma_3 \mathcal{L}_{\text{assoc}} + \gamma_4 \mathcal{L}_{\text{reg\_id}}$$
$$+ \gamma_5 \mathcal{L}_{\text{reg\_h\_orth}} + \gamma_6 \mathcal{L}_{\text{reg\_h\_sparse}}). \tag{4}$$

## 5.3 Learned data-independent symmetries

In the first set of tasks, we see if our method can learn generators for architectures with already known data-independent symmetries. We consider two-layer networks in the form of $L(W_1, W_2, X, Y) = \|W_2 \sigma(W_1 X) - Y\|^2$, where $W_2 \in \mathbb{R}^{m \times h}$, $W_1 \in \mathbb{R}^{h \times n}$ are parameters, $X \in \mathbb{R}^{n \times k}$, $Y \in \mathbb{R}^{m \times k}$ are data, and $\sigma$ is a homogeneous activation function.

During training, we train the generators $h$ and the group action $a$ under objective (4). We parametrize $a$ using a 4-layer MLP with hidden dimensions 128, 128, 128. The group aciton $a$ takes a group element, parameter, and data as input and outputs transformed parameters. We use 2000 training samples, each containing a randomly generated set of parameters and data. We set the learning rate as $10^{-3}$ and the weights for the multi-objective loss as $\gamma_1 = 10$, $\gamma_2 = \gamma_4 = \gamma_5 = 1$, and $\gamma_6 = 0.1$.

As a proof of concept, we train for a group action and a single generator $h \in \mathbb{R}^{2 \times 2}$ for the two-layer architecture with $m = h = n = k = 1$ and $\sigma$ being the identity function. Figure 3(a) presents the learned generator $h$ as a $2 \times 2$ matrix, which matches the expected generator that generates the rescaling group. Figure 3(b) visualizes a group element generated by $h$, corresponding to a rescaling group element.

Note that, however, we do not impose constraints on the group action (in particular, not enforcing linear actions). Hence we do not expect the learned generators to look similar to the elements of the Lie algebra infinitesimal generators of the symmetry group in general. For example, the action $a$ can be a composition of two function, the first transforming learned generators to the set of actual generators, and the second performing the group action. We find that our method can learn the generators and group actions for wider two-layer homogeneous architectures as well. More examples of learned generators for larger architectures can be found in Appendix C.

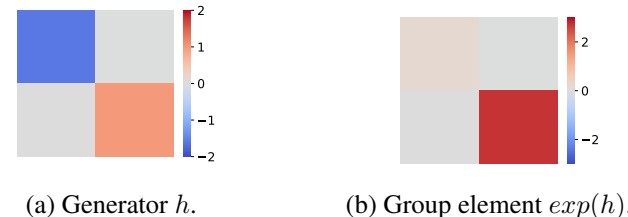

(a) Generator $h$.      (b) Group element $exp(h)$.

Figure 3: Learned generator for a two-layer linear MLP with scalar parameters and data, and a group element obtained via the exponential map.

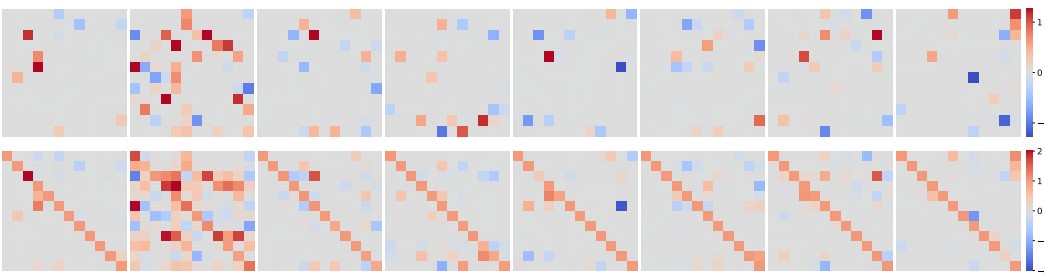

Figure 4: Learned data-dependent symmetries in a two-layer sigmoid MLP with parameters dimensions $W_2 \in \mathbb{R}^{3 \times 1}, W_1 \in \mathbb{R}^{3 \times 3}$ and data $X \in \mathbb{R}^{1 \times 3}, Y \in \mathbb{R}^{1 \times 1}$. Top: learned generators. Bottom: group elements obtained via the exponential map.

## 5.4 LEARNED DATA-DEPENDENT SYMMETRIES

As a more practical application of our framework, we attempt to uncover data-dependent symmetries from architectures where no continuous symmetry is known before. We apply our framework to learn generators and loss-invariant group actions for two-layer neural network with sigmoid and tanh activation function, as well as a three-layer neural network with skip connection.

Specifically, we aim to learn symmetries in the two-layer networks defined in the previous section, but replacing $\sigma$ by sigmoid or tanh. Our objective is again to find a set of generators $h$ and a group action $a$ that minimizes (4). We use 10000 training samples, each containing a randomly generated set of parameters and data. We set the learning rate as $10^{-3}$ and the weights for the multi-objective loss as $\gamma_1 = 1, \gamma_2 = \gamma_4 = 10, \gamma_5 = 1$, and $\gamma_6 = 0.1$.

Figure 4 shows the learned generators for data-dependent symmetries in a two-layer sigmoid MLP with parameters dimensions $W_1 \in \mathbb{R}^{3 \times 3}, W_2 \in \mathbb{R}^{3 \times 1}$ and data $X \in \mathbb{R}^{3 \times 1}, Y \in \mathbb{R}^{1 \times 1}$. Sigmoid networks have no data-independent continuous symmetry, and we observed that the derivative of the learned group action with respect to data is indeed nonzero. Therefore, this set of symmetries are data-dependent, indicating that our method successfully learns data-dependent symmetries for this architecture.

Figure 5 shows the learned generators for data-dependent symmetries in a three-layer tanh MLP with parameters dimensions $W_1 \in \mathbb{R}^{2 \times 2}, W_2 \in \mathbb{R}^{2 \times 2}, W_3 \in \mathbb{R}^{2 \times 1}$ and data $X \in \mathbb{R}^{1 \times 2}, Y \in \mathbb{R}^{1 \times 1}$. The generators indicate the existence of symmetries that act on non-contiguous layers, which has not been discovered in previous literature.

Although the loss invariance is only enforced locally during learning, the learned generators approximately preserves loss over the group actions using group elements from these generators. To verify loss invariance, we plot the loss on curves generated from the learned symmetries (Figure 6). Specifically, we compute the loss value on curves $\gamma(t) = a_X(\exp(ht), \theta)$ in $\Theta$, where $h$ is a learned Lie algebra element and $a_X$ is the learned group action. Compared to random curves generated by random Lie algebra elements $h$, the loss is lower on curves generated by the learned symmetries.

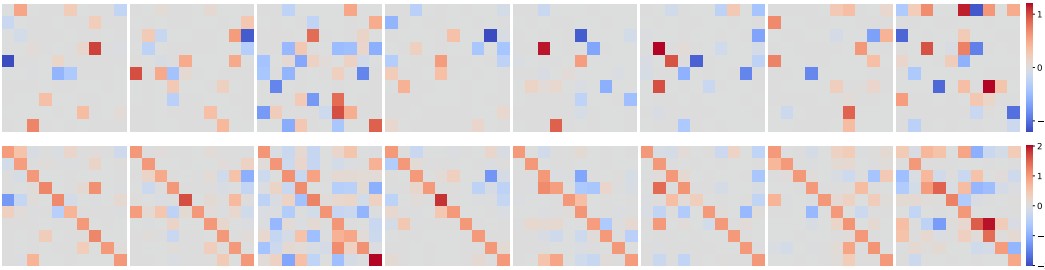

Figure 5: Learned data-dependent symmetries in a three-layer tanh MLP with parameters dimensions $W_1 \in \mathbb{R}^{2 \times 2}, W_2 \in \mathbb{R}^{2 \times 2}, W_3 \in \mathbb{R}^{2 \times 1}$ and data $X \in \mathbb{R}^{1 \times 2}, Y \in \mathbb{R}^{1 \times 1}$. Top: learned generators. Bottom: group elements obtained via the exponential map.

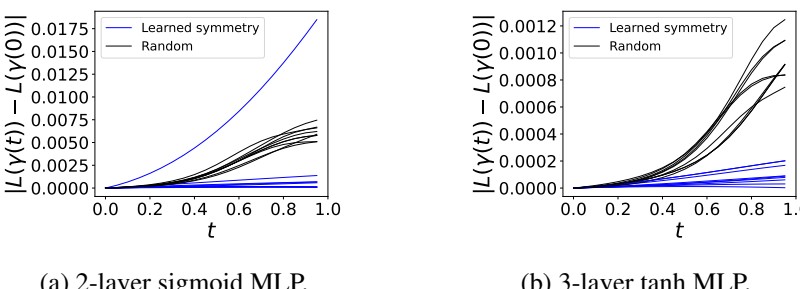

(a) 2-layer sigmoid MLP.                (b) 3-layer tanh MLP.

Figure 6: Loss on curves generated by learned symmetries vs. loss on random curves in the parameter space.

## 6 DISCUSSION

In this work, we introduce a framework for formalizing and discovering parameter space symmetries in neural networks. By deriving infinitesimal forms of these symmetries, we enable automated identification of loss-invariant transformations, including previously unknown ones. Additionally, we generalizes symmetries from smaller subnetworks to larger architectures, reducing computational complexity while improving scalability. These findings provide a foundational step toward a deeper understanding of data-dependent parameter space symmetries, uncovering new structural insights that could reshape how we analyze and optimize neural networks.

While our discovery framework suggests that there are previously unknown data-dependent symmetries in various neural network architectures, the existence and number of symmetries in neural network parameter spaces remain open questions. Whether the number of symmetries is affected by existence of symmetry in data or changes during training are also interesting directions. Future work will examine the structure of learned symmetry, such as the dimension of Lie algebras.

The discovered symmetries could be useful in downstream applications such as weight space learning. When the parameters of a neural network are used as input to make predictions, parameter space symmetry of the input network becomes the data space symmetry of processing network. Architectures that can respect the symmetries in the input networks has proven to be able to process neural network weights effectively (Zhang et al., 2023; Lim et al., 2024; Kalogeropoulos et al., 2024; Tran et al., 2024). Leveraging the learned symmetries via equivariant architectures (Finzi et al., 2021) or frame averaging (Puny et al., 2021) can potentially improve current approaches that are limited to known symmetries.

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

# APPENDIX

## A  INFINITESIMAL SYMMETRY AND EXAMPLES

### A.1  INFINITESIMAL FORMULATION FOR LOSS INVARIANCE

**Theorem 3.1.** *Let $a : \mathcal{D}^d \to (G \times \Theta \to \Theta)$ be a parameter space symmetry of a loss function $L : \Theta \times \mathcal{D}^d \to \mathbb{R}^d$. Assume that $L$ and $a_X$ is differentiable at all $X \in \mathcal{D}^d$. Let $D_\theta L|_{\theta,X} \in \mathbb{R}^{d \times \dim(\Theta)}$ be the derivative of $L$ with respect to $\theta$, and $D_g a_X|_{I,\theta} \in \mathbb{R}^{\dim(\Theta) \times \dim \mathfrak{g}}$ be the derivative of $a_X(g, \theta)$ with respect to $g$. Then, for all $\theta \in \Theta$, $X \in \mathcal{D}^d$, and $h \in \mathfrak{g}$,*

$$D_\theta L|_{\theta,X} \ D_g a_X|_{I,\theta}(h) = 0.$$

*Proof.* Since $a$ is a symmetry of $L$, we have

$$L(a_X(g, \theta), X) = L(\theta, X), \quad \forall g \in G, \quad \forall \theta \in \Theta, \quad \forall X \in \mathcal{D}^d.$$

Consider a smooth curve $\gamma(t) = a_X(\exp(ht), \theta)$ in $\Theta$, where $h \in \mathfrak{g}$ and $t \in \mathbb{R}$. Then, since $L$ is invariant under $a$,

$$L(\gamma(t), X) = L(\theta, X), \quad \forall t \in \mathbb{R}.$$

Differentiating both sides with respect to $t$ at $t = 0$, we get

$$\frac{d}{dt} L(\gamma(t), X) \bigg|_{t=0} = 0.$$

Applying the chain rule and noting that $\gamma(t)|_{t=0} = \theta$,

$$\frac{d}{dt} L(\gamma(t), X) \bigg|_{t=0} = D_\theta L|_{\theta,X} \left( \frac{d\gamma(t)}{dt} \bigg|_{t=0} \right).$$

Now, compute $\left.\frac{d\gamma(t)}{dt}\right|_{t=0}$ using the chain rule:

$$\left.\frac{d\gamma(t)}{dt}\right|_{t=0} = \left.\frac{d}{dt}a_X(\exp(ht),\theta)\right|_{t=0} = D_g a_X|_{I,\theta}\left(\left.\frac{d}{dt}\exp(ht)\right|_{t=0}\right).$$

Since $\exp$ is the exponential map from $\mathfrak{gl}(n)$ to $GL(n)$, and $h \in \mathfrak{gl}(n)$, we have

$$\left.\frac{d}{dt}\exp(ht)\right|_{t=0} = h.$$

Therefore,

$$\left.\frac{d\gamma(t)}{dt}\right|_{t=0} = D_g a_X|_{I,\theta}(h).$$

Putting it all together,

$$D_\theta L|_{\theta,X}\big(D_g a_X|_{I,\theta}(h)\big) = 0.$$

$\square$

### A.2 INFINITESIMAL FORMULATION FOR THE ASSOCIATIVITY AXIOM

Assuming the group action $a$ is smooth, we rewrite the associativity axiom,

$$a_X(g_2, a_X(g_1,\theta)) = a_X(g_2 g_1,\theta),$$

into an infinitesimal form that uses Lie algebras.

**Theorem 3.2.** *Let $G$ be a Lie group, and let $a : \mathcal{D}^d \to (G \times \Theta \to \Theta)$ be a smooth map that satisfies the associative law. Then, for all $\theta \in \Theta$, $X \in \mathcal{D}^d$, and $h_1, h_2 \in \mathfrak{g}$,*

$$D_\theta a_X\big|_{I,\theta}\big(D_g a_X\big|_{I,\theta}(h_1)\big) + D_g a_X\big|_{I,\theta}(h_2) = D_g a_X\big|_{I,\theta}(h_1 + h_2).$$

*Proof.* To capture infinitesimal behavior, we write $g_1 = \exp(th_1)$ and $g_2 = \exp(th_2)$, where $h_1, h_2 \in \mathfrak{g}$, and $t \in \mathbb{R}$. We then take the derivative of both sides of the associativity axiom with respect to $t$, at $t = 0$.

For the left side:

$$\frac{d}{dt}a_X(g_2, a_X(g_1,\theta)) = \frac{\partial}{\partial g_1}a_X(g_2, a_X(g_1,\theta))\frac{\partial g_1}{\partial t} + \frac{\partial}{\partial g_2}a_X(g_2, a_X(g_1,\theta))\frac{\partial g_2}{\partial t}$$

$$= D_\theta a_X\big|_{I,\theta}\big(D_g a_X\big|_{I,\theta}(h_1)\big) + D_g a_X\big|_{I,\theta}(h_2).$$

Following from the Baker-Campbell-Hausdorff formula in the first-order approximation, $g_2 g_1 \approx \exp(t(h_1 + h_2))$ for small $t$. For the right side of the associativity axiom:

$$\frac{d}{dt}a_X(g_2 g_1,\theta) = D_g a_X\big|_{I,\theta}(h_1 + h_2).$$

Both sides are equal in the associativity axiom. Therefore, their derivatives are equal. That is,

$$D_\theta a_X\big|_{I,\theta}\big(D_g a_X\big|_{I,\theta}(h_1)\big) + D_g a_X\big|_{I,\theta}(h_2) = D_g a_X\big|_{I,\theta}(h_1 + h_2).$$

$\square$

## B BUILDING SYMMETRIES FROM KNOWN ONES

This section contains the proofs for results in Section 4.

**Proposition 4.1.** *Let $L : \Theta \times \mathcal{D}^d \to \mathbb{R}^d$ be a function, where the parameter space $\Theta$ is a product space $\Theta = \Theta_1 \times \Theta_2$, with spaces $\Theta_1, \Theta_2$. Suppose there exist functions $h : \Theta_1 \times \mathcal{D}^d \to S$, $f : \Theta_2 \times S \to T$, and $j : (\Theta_1 \times T) \times \mathcal{D}^d \to \mathbb{R}^d$, such that for every $\theta = (\theta_1, \theta_2) \in \Theta$ and $X \in \mathcal{D}^d$, $L(\theta, X) = j\big((\theta_1, f(\theta_2, h(\theta_1, X))), X\big)$. If $a : S \to (G \times \Theta_2 \to \Theta_2)$ is a $G$-symmetry of $f$, then there is an induced $G$-symmetry of $L$, $a' : \mathcal{D}^d \to (G \times \Theta \to \Theta)$, defined by $a'_X(g, (\theta_1, \theta_2)) = \big(\theta_1, a_{h(\theta_1, X)}(g, \theta_2)\big)$.*

*Proof.* We need to show that $a'$ satisfies the identity and associative law of a group action and preserves $L$.

Since $a$ is a group action on $\Theta_2$, it satisfies the identity axiom $a_{h(\theta_1, X)}(I, \theta_2) = \theta_2$. Applying this in the definition of $a'$, we get $a'_X(I, (\theta_1, \theta_2)) = (\theta_1, a_{h(\theta_1, X)}(I, \theta_2)) = (\theta_1, \theta_2)$.

Since $a$ is a group action on $\Theta_2$, it satisfies the associative law $a_{h(\theta_1, X)}(g_2 g_1, \theta_2) = a_{h(\theta_1, X)}(g_2, a_{h(\theta_1, X)}(g_1, \theta_2))$, for all $g_1, g_2 \in G$. It follows that $a'$ also satisfies the associative law: $a'_X(g_2 g_1, (\theta_1, \theta_2)) = (\theta_1, a_{h(\theta_1, X)}(g_2 g_1, \theta_2)) = (\theta_1, a_{h(\theta_1, X)}(g_2, a_{h(\theta_1, X)}(g_1, \theta_2))) = a'_X(g_2, a'_X(g_1, (\theta_1, \theta_2)))$

Finally, since $a$ is a symmetry of $f$, we have $f(a_{h(\theta_1, X)}(g, \theta_2), h(\theta_1, X)) = f(\theta_2, h(\theta_1, X))$, for all $g \in G$. It follows that $a'$ preserves the value of $L$: $L(a'_X(g, \theta), X) = j\big((\theta_1, f(a_{h(\theta_1, X)}(g, \theta_2), h(\theta_1, X))), X\big) = j\big((\theta_1, f(\theta_2, h(\theta_1, X))), X\big) = L(\theta, X)$. $\square$

**Corollary 4.2.** *Consider a network parameter space $\Theta(m, h, n) = \mathbb{R}^{m \times h} \times \mathbb{R}^{h \times n}$ and data space $\mathcal{D}(n, k) = \mathbb{R}^{n \times k}$. Let $\sigma : \mathbb{R}^{h \times k} \to \mathbb{R}^{h \times k}$ be a row-wise function. Consider a function $L_{mnhk} : \Theta(m, h, n) \times \mathcal{D}(n, k) \to \mathbb{R}^{m \times k}$, defined as $L_{mnhk}((U, V), X) = U\sigma(VX)$ for $U \in \mathbb{R}^{m \times h}$, $V \in \mathbb{R}^{h \times n}$, and $X \in \mathbb{R}^{n \times k}$. If there is a $G$-symmetry of $L_{mnhk}$, then there is a $G$-symmetry of $L_{mnh'k}$ with any $h' > h$.*

*Proof.* The function $L_{mnh'k}$ can be decomposed into

$$U(\sigma(VX))_{ik} = U_{ij}\sigma(VX)_{jk}$$

$$= \sum_{j=1}^{h} \sum_{l=1}^{n} U_{ij}\sigma(V_{jl}X_{lk})$$

$$= \sum_{j=1}^{h} \sum_{l=1}^{n} U_{ij}\sigma(V_{jl}X_{lk}) + \sum_{j=h+1}^{h'} \sum_{l=1}^{n} U_{ij}\sigma(V_{jl}X_{lk}) \quad (5)$$

Note that for all $i, k$, the first term depends only on the first $h$ columns of $U$ and first $h$ rows of $V$, and the second terms depends only on the rest of the columns and rows of $U$ and $V$. Denoting the first $h$ columns of $U$ as $U_{1:h}$, the rest of the columns of $U$ as $U_{h+1:h'}$, the first $h$ rows of $V$ as $V_{1:h}$, and the rest of the rows of $V$ as $V_{h+1:h'}$, we have

$$L_{mnh'k}((U, V), X) = L_{mnhk}((U_{1:h}, V_{1:h}), X) + L_{mn(h'-h)k}((U_{h+1:h'}, V_{h+1:h'}), X). \quad (6)$$

Let $\Theta_1 = \mathbb{R}^{m \times h} \times \mathbb{R}^{h \times n}$ and $\Theta_2 = \mathbb{R}^{m \times (h'-h)} \times \mathbb{R}^{(h'-h) \times n}$. Then $\Theta(m, h', n) = \Theta_1 \times \Theta_2$. Let $S = (\mathbb{R}^{m \times k} \times \mathcal{D}^d)$ and $T = \mathbb{R}^{m \times k} \times \mathbb{R}^{m \times k}$. Define the following three functions

$$h : \Theta_1 \times \mathcal{D}^d \to (\mathbb{R}^{m \times k} \times \mathcal{D}^d)$$
$$f : \Theta_2 \times (\mathbb{R}^{m \times k} \times \mathcal{D}^d) \to \mathbb{R}^{m \times k} \times \mathbb{R}^{m \times k}$$
$$j : (\Theta_1 \times (\mathbb{R}^{m \times k} \times \mathbb{R}^{m \times k})) \times \mathcal{D}^d \to \mathbb{R}^{m \times k} \quad (7)$$

by

$$h((U_{1:h}, V_{1:h}), X) = (L_{mnhk}((U_{1:h}, V_{1:h}), X), X)$$
$$f((U_{h+1:h'}, V_{h+1:h'}), (Y, X)) = (L_{mn(h'-h)k}((U_{h+1:h'}, V_{h+1:h'}), X), Y)$$
$$j\big(((U_{1:h}, V_{1:h}), (Y', Y)), X\big) = Y' + Y. \quad (8)$$

Then $L_{mnh'k}(\theta, X) = j\big((\theta_1, f(\theta_2, h(\theta_1, X))), X\big)$ for all $\theta = (\theta_1, \theta_2) \in \Theta$ and $X \in \mathcal{D}^d$. Since $L_{mnhk}$ has a symmetry, $f$ has the same symmetry. By Proposition 4.1, $L_{mnh'k}$ also has the same symmetry. $\square$

**Corollary 4.3.** *Let $\Theta = \Theta_1 \times ... \times \Theta_l$ be a parameter space. Consider a list of spaces $V_0 = \mathcal{D}^d$, $V_l = \mathbb{R}^d$, and $V_1, ..., V_{l-1}$. Let $L : \Theta \times \mathcal{D}^d \to \mathbb{R}^d$ be a function defined recursively by $\{L_i\}_{i=1}^l$ with $L_i : \Theta_i \times V_{i-1} \to V_i$, such that $L = \phi_l$ where $\phi_i = L_i(\theta_i, \phi_{i-1}) \in V_i$ and $\phi_0 = X$. If for some $1 \leq i \leq l$, $L_i$ has a $G$-symmetry, then $L$ has a $G$-symmetry.*

*Proof.* Define functions

$$h : (\Theta_1 \times ... \times \Theta_{i-1} \times \Theta_{i+1} \times ... \times \Theta_l) \times \mathcal{D}^d \to V_{i-1}$$
$$f : \Theta_i \times V_{i-1} \to V_i$$
$$j : (\Theta_1 \times ... \times \Theta_{i-1} \times \Theta_{i+1} \times ... \times \Theta_l) \times V_i \times \mathcal{D}^d \to \mathbb{R}^d \quad (9)$$

by

$$h((\theta_1, ..., \theta_{i-1}, \theta_{i+1}, ..., \theta_l), X) = L_{i-1}(\theta_{i-1}, X), \quad \text{computed using } (\theta_1, ..., \theta_{i-1})$$
$$f(\theta_i, \phi_{i-1}) = L_i(\theta_i, \phi_{i-1})$$
$$j((\theta_1, ..., \theta_{i-1}, \theta_{i+1}, ..., \theta_l), \phi_i, X) = L_l(\theta_l, X), \quad \text{computed using } (\theta_l, ..., \theta_{i+1}) \text{ and } \phi_i. \quad (10)$$

Then $L((\theta_1, ..., \theta_l), X) = j\big((\theta_1, ..., \theta_{i-1}, \theta_{i+1}, ..., \theta_l), f(\theta_i, h((\theta_1, ..., \theta_{i-1}, \theta_{i+1}, ..., \theta_l), X)), X\big)$ for all $\theta = (\theta_1, \theta_2) \in \Theta$ and $X \in \mathcal{D}^d$. By Proposition 4.1, if $f = L_i$ has a $G$-symmetry, $L$ also has a $G$-symmetry. $\square$

**Proposition 4.4.** *Let $L_d : \Theta \times \mathcal{D}^d \to \mathbb{R}^d$ be a function that is applied pointwise on each of $d$ data points in a data batch. If $L_d$ admits a $G$-symmetry, then $L_{d'}$ admits a $G$-symmetry for all $d' < d$.*

*Proof.* Suppose that $L_d$ has a $G$-symmetry. Let $a : \mathcal{D}^d \to (G \times \Theta \to \Theta), X_d \mapsto (a_{X_d} : g, \theta \mapsto \theta')$ be the corresponding group action. Define $a' : \mathcal{D}^{d'} \to (G \times \Theta \to \Theta)$ by $X_{d'} \mapsto (a_{t(X_{d'})} : g, \theta \mapsto \theta')$, where $t : \mathcal{D}^{d'} \to \mathcal{D}^d$ appends $d - d'$ random data points to its input. Clearly, $a'$ satisfies the identity and associate axiom and preserves loss. Therefore, $a'$ is a $G$-symmetry of $L_{d'}$. $\square$

## C  ADDITIONAL EXPERIMENT DETAILS

### C.1  ALTERNATIVE OPTION FOR DISCOVERY OBJECTIVES

A more straightforward training objective exponentiates the Lie algebra to obtain group elements, before enforcing loss invariance and group axioms:

$$\min_{h,a} L_{\text{invariance\_int}} + L_{\text{id\_int}} + L_{\text{assoc\_int}}$$

with

$$L_{\text{invariance\_int}} = \mathbb{E}_{x,\theta,t} \| L(a_x(exp(ht), \theta), x) - L(\theta, x) \|$$
$$L_{\text{id\_int}} = \mathbb{E}_{x,\theta} \| a_x(I, \theta) - \theta \|$$

$$L_{\text{assoc\_int}} = \sum_{h_1, h_2 \in \mathfrak{g}} \mathbb{E}_{x,\theta} \left\| a_{\exp(h_1)X}(\exp(h_2), a_X(\exp(h_1), \theta)) - a_X(\exp(h_2)\exp(h_1), \theta) \right\|.$$

Similarly to the infinitesimal version, this objective also directly enforces the necessary group structures. We adopt the infinitesimal formulation to avoid the computational overhead of evaluating exponential maps.

### C.2  ADDITIONAL RESULTS

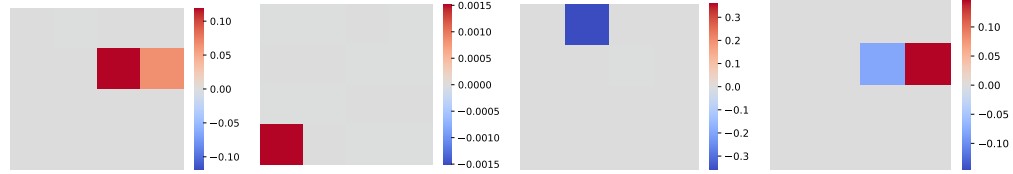

Figure 7: Learned generators for a two-layer linear MLP with parameters dimensions $W_2 \in \mathbb{R}^{1\times2}, W_1 \in \mathbb{R}^{2\times1}$ and data $X, Y \in \mathbb{R}$.

