# OpenReview forum: "Finding Symmetry in Neural Network Parameter Spaces"
_ICLR.cc/2026/Conference — Submitted to ICLR 2026_

### Official Review · Reviewer_Sfv7 · 2025-10-24

**Soundness:** 3
**Presentation:** 3
**Contribution:** 2
**Rating:** 6
**Confidence:** 2

**Summary:**

The authors claim to make a few contributions to finding symmetries in neural network parameter spaces. They claim to introduce an automated approach for discovering symmetry groups (and their group actions) in these parameter spaces, by defining the search space and using neural networks to learn them. They also claim that they can use discoveries about the symmetries of parameter spaces for smaller neural networks to learn about those for larger ones where the smaller ones are a subnetwork of the larger ones.

**Strengths:**

Let me state clearly for the record that the topic area is not one I am overly familiar with (i.e. I do not know the prior literature, nor - prior to reading this paper - the central questions in this topic area). So I suggest that my review is taken with a pinch of salt.

In saying that, I thought that
- the paper was very well written and I could get a sense of the problem that they were tackling, as well as why one would want to look at this problem
- the proofs of their theorems are correct (to the best of my knowledge)
- the authors demonstrate what they set out to achieve

**Weaknesses:**

I only have one particular thought on this, which is:

How useful is it just to prove the existence of certain symmetries? I understand that the authors have demonstrated the existence of certain ones that weren't known before using their framework, but how would one go on and use this knowledge in the potential applications that they list in their introduction (such as optimising neural networks)?

**Questions:**

Beyond the question posed in the weaknesses section: I would be interested in the authors commenting on the following:

Can the results be extended beyond linear groups? If not, what are the key issues behind this?

(Also I think there may be a mistake in line 325, do they not mean L_{invariance} instead of L_{Lie_deriv}?)

---

### Official Review · Reviewer_vVw6 · 2025-10-29

**Soundness:** 2
**Presentation:** 3
**Contribution:** 2
**Rating:** 2
**Confidence:** 5

**Summary:**

The authors study and provide methods for discovering symmetries in neural networks parameter spaces. Focusing on infinite groups and their Lie generators, they have the following contributions. First, in Theorem 3.1 and 3.2 they prove the necessary conditions that the loss and group action derivatives have to obey for a given set of generators. This allows them to promote these quantities during training, yielding the proposed loss function in Equation 4. Moreover, in Section 4 they proposed a modular viewpoint of symmetry discovery in neural networks parameter spaces and show that symmetries of small subnetworks of a big model will eventually appear as symmetries of the whole model. This is characterized in Proposition 4.1 and is applied to neural networks in Corollaries 4.2 and 4.3. They complete this section by proving that data-dependent symmetries, if observed in a batch, are also valid symmetries for subsets of the batch. The paper is concluded with a number of experiments (small scale) to show that for both data-dependent and data-independent symmetries, their proposed loss provides non-trivial solutions. The experiments are mostly providing a proof of concept for the proposed loss.

**Strengths:**

- Neural network parameter spaces and their symmetries are an active area of research, and results in this direction are potentially interesting to a large body of the community.

- The paper is well written so that it needs minimal background to understand the method and results.

**Weaknesses:**

- The experiments are just for the proof of the concept. However, the main contribution of this paper is the introduction of a loss that promotes symmetry discovery and allows to algorithmically achieve it. This work needs large-scale, comprehensive experiments before publication, in my opinion.

- The applications of discovering data-dependent symmetries are missing. One expects at least some experiments and some toy theory on why we need such things.

**Questions:**

This is a nice, well-written paper on how to discover symmetries in neural networks' parameter spaces. The paper's most important contribution is a new loss function in Equation 4, along with a number of theoretical results, which provide an algorithm for symmetry detection. The focus of the paper is on infinite groups, and data depedendant symmetries are also handled in this framework.


However, I think this paper at this moment is not ready for publication. The main reason is that the main contribution is the introduction of a method, a loss function, which leads to an algorithm for symmetry discovery. But the authors just provided proof of concept, small-scale experiments, which are not enough to validate the method for real cases of deep learning. One needs large scale (or at least mid scale) experiments, with concrete applications. Indeed, why are such symmetries in data needed to be discovered? What kind of underlying task gets resolved in I learned data-dependent symmetries? Are data-dependent symmetries some kind of noisy signals discovered from the overparametrized models, or do they carry a meaningful signal? In my opinion, the paper needs to have an explicit experiment showing that the application of the data-dependent symmetry discovery leads to some performance improvement. One may find bio-related datasets useful in this direction.

I feel my concerns need some time well beyond the rebuttal period to be applied. So at the moment I recommend rejection, while encouraging the authors to apply the comments above and have a completely nice paper.

Some questions:

 - How does the objective in Section 5.1 enforce group law? Any proofs? Or this is just heuristics?

 - Section 5.2: How is the hyperparameter beta chosen? Grid search, or there is some formula that at least gives us some hint on where to search?

 - Why Equation (4) returns a good symmetry? Any proofs?

 - Figure 4: An interpretation of what those data-dependent symmetries mean is missing



 - What does Figure 2 mean? It's a bit unclear (I understand what it means, but it takes some time so it's not providing much help for the reader)


 - Line 296: typo: 'a effective'

---

### Official Review · Reviewer_rWxR · 2025-10-30

**Soundness:** 3
**Presentation:** 2
**Contribution:** 2
**Rating:** 4
**Confidence:** 3

**Summary:**

This paper considers the problem of finding data-dependent symmetries in neural network parameters. Recall that parameter symmetries are typically defined as transformations that can be performed on a neural network of arbitrary weights which result in an equivalent function (different weights, same function). This paper considers the broader question of data-dependent transformations (that is, the transformations only result in the same function on a subset of datapoints). It starts out by describing a formal framework of definitions to make this precise. It then translates these definitions into an infinitesimal version that can be calculated at the level of the Lie algebra, providing a couple of examples along the way. This is important since all calculations in practice will be performed at this local level. Finally, the paper describes an empirical method for finding symmetries by learning Lie algebra element actions on network parameters. Several very small-scale experiments ($<25$ network parameters) are analyzed.

**Strengths:**

**An interesting and important topic, the research direction seems promising:** It is this reviewer’s opinion that neural network symmetries remain an underappreciated topic in deep learning (though this seems to be steadily changing). Further, since many symmetries that are easy for a human to identify have probably already been found, turning to data-driven approaches seems like a promising direction.

**The paper presents a decent foundation on which to build the proposed method:** While the reviewer has comments in the *Weaknesses* section about presentation and organization, he did appreciate the way in which the paper lays out all of the supporting  material before presenting the main method in the work. The reviewer thinks this is important. Too many works in this field rush to present new methods/results without establishing notation and definitions (which can shift from work to work), leading to confusion and misinterpretation. With some polishing this paper could read very well.

**Focusing on some small-scale examples is helpful:** This reviewer anticipates that there may be some criticism of this paper for performing experiments on very small networks. While the reviewer agrees that including larger-scale experiments would ultimately make the paper stronger, he also thinks the small-scale experiments are an important component. Many readers of this type of work will likely want to understand new symmetries, not just know they are there. This understanding probably requires working with small examples.

**Weaknesses:**

**Deeper analysis of the experiments would be valuable:** The reviewer is very interested in this direction of research and was disappointed that the analysis of the experimental results was cursory. One can imagine a very thorough analysis that could help point the way towards very specific, non-trivial symmetry discovery (that could be later verified formally). This would be a great way to validate the method and would add to the scientific contribution. It would also align with the current experimental set-up which focuses on small networks.

**The motivation for data-dependence was not clear:** Reading the first half of this paper, it felt like the motivation was to look for symmetries that were specifically data dependent. At the end however, it seemed that data-dependence may have been included because the method for finding symmetries depends on evaluation on a finite set of datapoints (and hence is data-dependent by necessity). Which is it? If data-dependence is introduced because the method requires it, that is fine, but it would be helpful to state this up-front.

**More experiments:** Along with deeper analysis, it would be helpful to have more experiments (even if they remain small scale). The reviewer is curious what kinds of groups end up showing up here? Is there an easy way to measure this? It seems that some of the familiar symmetries (e.g., permutations) don’t appear because they are coming from finite groups?


**Nitpicks:**
- Section 3.1: It might be helpful to put in a single example or two to help ground this section. Maybe a case the reader will be familiar with.
- Line 139: May be worth reminding the reader you are only considering Lie groups here.
- Line 354: It may just be this reviewer’s opinion, but underscores look better in code variables than in LaTeX.
- Line 325: I don’t think $\mathcal{L}_{Lie deriv}$ is defined?
- Line 119: Is ‘associative’ the right word here? This seems to be essentially the homomorphism property?
- Line 187: ‘To facilitates…’ $\mapsto$ ‘To facilitate…’.
- Line 336: “Therefore, in implementation…” $\mapsto$ “Therefore, in our implementation…”

**Questions:**

- Line 112: The reviewer is a little confused about this definition. Informally, a set $X$ from the data space $\mathcal{D}$ is sent to a map $a_X$ which takes pairs $(g,\theta)$ and applies $g$ to $\theta$ to get $\theta’$. Where does $X$ come in? There is just some arbitrary dependence on $X$ that is specified case by case?
- Line 362: How do you choose the number of $h$?

---

### Official Review · Reviewer_Uz1c · 2025-11-01

**Soundness:** 2
**Presentation:** 2
**Contribution:** 1
**Rating:** 4
**Confidence:** 3

**Summary:**

The paper introduces an automated framework for discovering parameter space symmetries that keep a neural network’s loss function unchanged. It formalizes data-dependent symmetries, derives their infinitesimal form, and enables efficient learning of nonlinear and data-dependent group actions that can generalize from small subnetworks to larger architectures. Empirically, the method recovers known symmetries and uncovers new data-dependent and cross-layer ones, providing a principled step toward systematic symmetry discovery in neural networks.

**Strengths:**

The paper proposes a clear and well-founded framework for discovering parameter symmetries in neural networks by defining data-dependent and infinitesimal forms. The theory is mathematically sound and shows how symmetries in small subnetworks can extend to larger ones. The experiments, though simple, support the main idea by recovering known symmetries and suggesting new ones.

**Weaknesses:**

1. The paper proposes an automated discovery framework for finding parameter symmetries, but the experiments only visualize small toy models (2–3 layer MLPs with a few parameters). There are no quantitative metrics showing how well the learned symmetries actually preserve loss invariance or how robust they are compared to random directions. Without such quantitative evidence, it is unclear whether the discovered new symmetries are genuine or artifacts of optimization.

2. The claimed goal is to find symmetries in large neural networks by extending from smaller subnetworks, but the experiments never go beyond extremely small MLPs. The paper does not test on modern architectures like Transformers or ResNets, where the approach’s scalability and generality would matter most. Thus, it remains unproven that the framework can meaningfully handle high-dimensional parameter spaces.

3. While the authors claim to discover previously unknown data-dependent or non-contiguous symmetries, there is no rigorous verification or analytical characterization of these symmetries. The results are limited to visual matrices of learned generators, without mathematical confirmation that they correspond to valid group actions or new invariances.

**Questions:**

1. How sensitive is the discovery process to the choice of architecture for the action network $a_X$? Does the method require specific inductive biases or regularization strategies to ensure meaningful, interpretable group actions rather than degenerate solutions?

2. In the infinitesimal formulation, the framework assumes smoothness and differentiability of the loss with respect to parameters. How does this apply to piecewise-linear networks such as ReLU architectures, where the loss landscape is not globally smooth?

3. One of the paper’s stated motivations is improving downstream applications such as optimization and generalization. Have the authors attempted to use the discovered symmetries to modify training dynamics, e.g., via symmetry-aware optimization or reparameterization, and if so, what were the outcomes?

---

### Official Review · Reviewer_Cv5L · 2025-11-11

**Soundness:** 3
**Presentation:** 3
**Contribution:** 3
**Rating:** 6
**Confidence:** 3

**Summary:**

This paper provides a formalization of data-dependent parameter symmetries in NNs. It also proposes an automated framework to discover them, and runs experiments in some interesting toy settings. The discovery method relies on learning an infinitesimal form of the symmetry, optimizing an objective function (Eq. 4) that enforces loss invariance, the identity axiom, and associativity. They find some previously unknown data-dependent symmetries in architectures like sigmoid and tanh MLPs that e.g. act on non-contiguous layers.

**Strengths:**

- The paper provides a novel formalization of data-dependent symmetries, and a discovery algorithm using Lie algebras.
- The paper makes and validates several interesting testable predictions, like the existence of novel, data-dependent symmetries in non-homogeneous networks. The authors show that the loss remains nearly constant along the paths generated by these learned symmetries.

**Weaknesses:**

- I am confused about Thm. 3.2. The group action's identity axiom seems to force the term D_{\theta} a\_{X}|\_{I, \theta} to be the identity map. This would cause the theorem to collapse to a simple statement about linearity, rendering the \mathcal{L}\_{\text{assoc}} loss term "vacuous" and failing to impose any new constraint from associativity.
- There is limited empirical scope. The results are on small 2–3 layer MLPs, and it’s not clear how scaling to data-dependent symmetries in in larger architectures will go. Some discussion here seems merited.

**Questions:**

Some small issues with the loss term as it is written in the paper, that all should be easily patched:
- Orthogonality: The regularizer is described as enforcing orthogonality but is incorrectly formulated. By minimizing the cosine similarity, it actively encourages generators to become anti-parallel (cosine = -1) rather than orthogonal (cosine = 0).
- Regularization: The \mathcal{L}\_{\text {reg_id }} term contains a redundant inner $\min_{a,h}$, which would cause the term to collapse to a constant during optimization instead of acting as a regularizer.
- Associativity: The main \mathcal{L}\_{assoc} loss is introduced as a vector but is used as a scalar in the final objective function (Eq. 4) without specifying the required norm (e.g., $L\_2$ norm) needed to make it a scalar value.

---

### Author Response · Authors · 2025-12-04

We thank all reviewers for their careful reading and thoughtful feedback. We especially appreciate the critiques regarding the limited scope of experiments and the helpful suggestions on additional experiments. These include demonstrating our method on larger networks and different architectures, as well as more analysis of the discovered symmetries. We also appreciate the questions on technical aspects of the infinitesimal formulation and the loss terms, which we will address in a future version.

We are encouraged that the reviewers found our research direction interesting and like our theoretical results. With expanded experiments and clearer motivation for data-dependent symmetries, we believe we can present a much stronger version of this work.

---

### Meta-Review · Area_Chair_Fpnu · 2026-01-07

**Summary:**

The article studies data dependent symmetries in neural network parameters.

Reviewers find the formalisation and algorithm novel and sound. Critique includes lack of quantitative metrics, small scale experiments, and lack of rigorous characterisations. Authors did not provide a rebuttal, but provided an official comment stating that with the reviews they feel confident that they can present a stronger version of the work in the future.

In consideration of this, I must reject the article. I thank the authors for their honest comment and encourage them to pursue the revisions and resubmit.

**Reviewer Concerns:**

Reviewer vVw6:

The experiments are just for the proof of the concept.
-> No rebuttal / response

The applications of discovering data-dependent symmetries are missing.
-> No rebuttal / response

**Reviewer Scores:**

For each review, specify how you think the reviewer would have changed their score if they had been able to participate fully in the discussion.

Reviewer Cv5L: 6 -> 6
Reviewer Uz1c: 4 -> 4
Reviewer rWxR: 4 -> 4
Reviewer vVw6: 2 -> 2
Reviewer Sfv7: 6 -> 6

---

### Decision · Program_Chairs · 2026-01-26

Reject